# Reliability of COVID-19 data: An evaluation and reflection

April R. Miller[1]☯, Samin Charepoo[2]☯, Erik Yan[3]☯, Ryan W. Frost[4], Zachary J. Sturgeon[5], Grace Gibbon[6], Patrick N. Balius [7], Cedonia S. Thomas[8], Melanie A. Schmitt[9], Daniel A. Sass[10], James B. Walters[11], Tracy L. Flood[11], Thomas A. Schmitt[11]‡*, on behalf of the COVID-19 Data Project¶

1 Department of Public Health, Simmons University, Boston, Massachusetts, United States of America, 2 Department of Data Science and Neuroscience, Simmons University, Boston, Massachusetts, United States of America, 3 Duke Global Health Institute, Duke University, Durham, North Carolina, United States of America, 4 Department of Mathematics & Statistics, Boston University, Boston, Massachusetts, United States of America, 5 Department of Physical Sciences, University of California San Diego, La Jolla, California, United States of America, 6 Global School of Public Health, New York University, New York City, New York, United States of America, 7 Division of Environmental Health Sciences, University of Minnesota, Minneapolis, Minnesota, United States of America, 8 Department of Biology, Tougaloo College, Tougaloo College, Tougaloo, Mississippi, United States of America, 9 Pediatric Ophthalmology and Adult Strabismus, Department of Ophthalmology and Visual Sciences, University of Wisconsin-Madison, Madison, Wisconsin, United States of America, 10 Department of Management Science and Statistics, The University of Texas at San Antonio, San Antonio, Texas, United States of America, 11 BroadStreet Health, Milwaukee, Wisconsin, United States of America

☯ These authors contributed equally to this work.
‡ TAS worked as lead author to this work.
¶ All authors were members of (or participated in) the COVID-19 Data Project [31]
* tom@broadstreet.io

**Data Availability Statement:** All the data underlying the findings of the study are available at the provided URL (https://github.com/BroadStreet-Health/COVID-19-Cases-and-Mortalities).

## Abstract

### Importance

The rapid proliferation of COVID-19 has left governments scrambling, and several data aggregators are now assisting in the reporting of county cases and deaths. The different variables affecting reporting (e.g., time delays in reporting) necessitates a well-documented reliability study examining the data methods and discussion of possible causes of differences between aggregators.

### Objective

To statistically evaluate the reliability of COVID-19 data across aggregators using case fatality rate (CFR) estimates and reliability statistics.

### Design, setting, and participants

Cases and deaths were collected daily by volunteers via state and local health departments, as primary sources and newspaper reports, as secondary sources. In an effort to begin comparison for reliability statistical analysis, BroadStreet collected data from other COVID-19 aggregator sources, including USAFacts, Johns Hopkins University, New York Times, The COVID Tracking Project.

**Funding:** Funders for this study include only BroadStreet Health, which provided support in the form of salaries for authors EY, RWF, and ZJS, but did not have any additional role in the study design, data collection and analysis, decision to publish, or preparation of the manuscript.

**Competing interests:** The specific roles of these authors are articulated in the 'author contributions' section. This does not alter our adherence to PLOS ONE policies on sharing data and materials.

## Main outcomes and measures

COVID-19 cases and death counts at the county and state levels.

## Results

Lower levels of inter-rater agreement were observed across aggregators associated with the number of deaths, which manifested itself in state level Bayesian estimates of COVID-19 fatality rates.

## Conclusions and relevance

A national, publicly available data set is needed for current and future disease outbreaks and improved reliability in reporting.

## Introduction

In the wake of the COVID-19 (2019 novel coronavirus) pandemic, death rates and spatial mapping, not dissimilar to methods used during the 19th century London cholera outbreak, have become talking points of the 21st century [1, 2]. As COVID-19 slowly gained momentum in late winter and early spring of 2020, governments and other organizations scrambled to collect and present temporo-spatial data. When governments, understandably, struggled with the proliferation of COVID-19, many non-governmental organizations and universities helped with the COVID-19 data collection by innovating with data aggregation techniques (e.g., web scraping, crowd-sourcing) [1–4].

Despite new technology and methods, aggregating COVID-19 data remains difficult and potentially error-prone due to the sheer amount of data collection methods and definition disparities, the novelty of the worldwide data tracking process, and the attempt to collect community-level information (e.g., county) [5]. Due to the efforts of multiple aggregators and their various methods, the COVID-19 pandemic provides a unique opportunity to evaluate the statistical reliability of near-real time and fast-moving infectious disease surveillance data. Thus, a well-documented reliability study examining and understanding data collection methods and possible causes of differences between aggregators, as well as methods for correcting these differences, is essential knowledge for future infectious disease outbreaks. This is the current goal of this paper: to take a tiny glacial step and begin the evaluation of the COVID-19 data collection process.

Despite the validity challenges, such as relying on surveillance data, as Hartley and Perencevich [6] have pointed out, leveraging continuous COVID-19 data from multiple sources to evaluate public health interventions in near real-time far outweighs the inevitable inaccuracies. In fact, technology has truly transformed the disease surveillance response to COVID-19 [7, 8]. Local and federal governments, hospitals, newspaper outlets, universities, the Centers for Disease Control and Prevention (CDC) and other organizations have worked together to aggregate and survey COVID-19 [9]. Once collected and combined by aggregators, COVID-19 data has been used to create geo-maps and other data visualizations and statistical models to track, predict, and understand the virus [8, 10, 11]. This aggregated COVID-19 data has helped governments and communities formulate responses, allocate resources, measure the effectiveness of policy interventions, such as stay-at-home orders, mask mandates, vaccinations, and provide guidance in loosening restrictions [12–15].

As a result, the COVID-19 data collection and reporting process has transformed, not only our ability to surveil infectious disease, but to also forecast cases and outcomes under varying government policies [8, 11, 16–18]. This has been integral in preventing disease and saving lives [19]. While new data and technology solutions have been invaluable in easing uncertainty and providing facts in a sea of unknowns, much of the data still remains unavailable at finer resolutions than the county-level [1, 7]. There is also the challenge of aggregating data across government agencies that collect and report their data differently [20–23]. Information and transparency in the U.S. related to data tracking methods is often limited and varied [24] as there are numerous independently established systems for reporting disease cases and deaths [6, 9]. Thus, numerous challenges remain with collecting and aggregating reliable COVID-19 data [24, 25].

In an effort to help move "real-time" disease surveillance forward, this study, done by BroadStreet in conjunction with the COVID-19 Data Project [26], attempts to make several contributions. First, we describe our COVID-19 data collection process and observations of the process, which has not previously been documented for COVID-19. Second, we examine the reliability of several COVID-19 data aggregators, including the CDC-endorsed USAFacts (USAF) [27], Johns Hopkins University (JHU) [28], New York Times (NYT) [29], The COVID Tracking Project (CTP) [30], and BroadStreet (BS), [26] at the national, state, and county levels. These aggregators were chosen because their data collection methods and data sets are publicly available, and are cited and used by a variety of organizations (e.g., CDC and Google). Understanding the differences between these aggregators will allow researchers to focus on, and potentially develop, more reliable disease tracking and data collection methods. Lastly, it is essential to examine how COVID-19 reporting differences may be manifested in commonly used tracking statistics, thus, we examined the case fatality rate estimates at the state level.

## Methods

### Data collection process

Starting on March 16th, 2020, the Broadstreet team (consisting of approximately 120 volunteers) [31] began tracking diagnosed cumulative cases of, and deaths due to COVID-19 reported by state and county governments [26]. Broadstreet volunteers were recruited from a variety of universities through public health and other related undergraduate and graduate departments, and they were eligible to participate in this project if they had any interest or experience in a public health-related field. Following CDC guidelines published on 4/5/2020 [32], these volunteers tracked case and death totals using various sources and organizing them within Google Sheets. Volunteers were organized into six regional teams consisting of members acting in daily data entry, management, and quality assurance roles. Probable cases are defined by the CDC as being:

1. Diagnosed through epidemiologically linking individuals expressing COVID-19 symptoms to a known case;

2. Tests positive for presence of antigen, and either expresses COVID-19 symptoms or is epidemiologically linked to a known case; or,

3. Meeting vital records criteria [33]. All data prior to March 9th was entered from Johns Hopkins University [28] and validated through multiple methods. Between March 9th and March 16th of 2020, data was obtained retroactively from state, county, and news sources.

County-level case and death count totals were entered daily in a 24-hour cycle to track cumulative totals of the virus over time. The spreadsheet consisted of rows for each county so

COVID-19 totals could be entered, as well as an "Unknown County" for cases that could not be assigned to a county. The sources used were official state or local government websites. In some cases this was supplemented with secondary sources, such as news sources, due to infrequent or nonspecific reporting by primary sources. Team managers examined the accuracy of daily totals to identify and correct errors. The Quality Assurance team then compared Broad-Street's county-level cumulative totals to those reported by other aggregators, including the NYT [29], JHU [28], CTP [30], and USAF [26], to check the reliability of entered data. If significant discrepancies between aggregators existed, Quality Assurance performed research to determine the most "accurate" count totals, and then left a comment with the results of their research and any changes. All team members signed off on a tracking sheet after completing their assigned tasks to ensure accountability.

In situations where the decrease was caused by a one-day anomaly in the totals reported, this was assumed to be a reporting error and the anomalous data was updated to match the following day. In the case of a simple decrease in cumulative totals, if research did not produce an explanation of the cause, then the assumption was made that this was due to cases or deaths being reassigned to a different county, and the historic totals in the initial county were reduced and transferred into an "Unknown County".

In an effort to begin comparison for reliability statistical analysis, BroadStreet collected data from other COVID-19 aggregator sources, including USAF, NYT, JHU, and CTP [27–29]. Initial examples of differences in reporting included BroadStreet reporting 89 more counties when compared to other aggregators and thousands of anomalies (number is less than previous day) in county-level case and death counts. This result is highly biased against counties with rapidly growing cases and deaths and illustrates that reported numbers are not always immediately accurate. Table 1 provides a summary of various data collection methods by the different aggregators.

## Data cleaning and preparation

Each aggregator's original dataset was cleaned and pre-processed in the R statistical computing language [35] to generate comparable datasets across aggregators. This included the removal of unknown case or death data, removal of unmatched Federal Information Processing Standards (FIPS), and removal of uncommonly reported US territories and/or smaller geographic

**Table 1. Aggregator data collection methods.**

| Variable | | Aggregator | | | | |
|---|---|---|---|---|---|---|
| | | USAF [27] | JHU [34] | NYT [29] | CTP [30] | BS [26] |
| **Collection Method** | **Web Scrapped** | ✓ | ✓ | | | |
| | **Manual Entry** | ✓ | ✓ | ✓ | ✓ | ✓ |
| | **Crowd Sourced** | | ✓ | | | |
| **Source** | **State Health Department** | ✓ (Daily) | | ✓ | ✓ (Daily) | ✓ |
| | **County Health Department** | | ✓ | ✓ | ✓ | ✓ |
| | **Media** | | ✓ | ✓ | | ✓ |
| **Data Collected** | **Cases** | ✓ (Cumulative) | ✓ | ✓ (Cumulative) | ✓ | ✓ |
| | **Deaths** | ✓ (Cumulative) | ✓ | ✓ (Cumulative) | ✓ | ✓ |
| | **Testing** | | | | ✓ | |
| **Quality Assurance** | **Revise Errors** | ✓ | ✓ | | | ✓ |
| | **Validate on other aggregators** | | | | | ✓ |

[a] USAF = USA Facts; NYT = New York Times; JHU = John Hopkins University; BS = BroadStreet; CTP = The COVID Tracking Project

delineations. In an effort to reduce the negative binomial skew resulting from a significant number of zero-count days (i.e., before COVID-19 was prevalent in a location), a date range of March 15, 2020 through June 30, 2020 was selected to reduce this concern while maximizing the inclusion of early count data.

A daily case and death count was calculated using a date's cumulative count minus the preceding date's cumulative count. Negative counts resulting from daily count calculation for cases and deaths were dropped from the dataset. Negative counts accounted for 0.77% of county case data, 0.21% of county death data, 0.05% of state case data, and 0.15% of state death data. Before conducting the reliability analyses, two other notable changes were made to the data; daily counts were smoothed using a 3-day moving average to account for asynchronous reporting of daily cases by each aggregator and all daily case and death counts were modified by +1 to remove any remaining zero-count data to improve Cohen Kappa coefficient estimates and reduce the number of paradoxical results.

The issue of smoothing is significant for several reasons. First, data were extremely unreliable prior to smoothing due to the reporting process and significant outliers in the data, thus the smoothed results presented here are more positive (i.e., possess higher reliability estimates) from a data aggregator perspective. Second, this finding suggests that government officials and media agencies should utilize and stress moving averages (or some other form of smoothed data) rather than raw new counts given their lack of reliability, and therefore potentially varying fatality estimates, across data sources. This finding largely explains the significant increases and decreases in counts that are commonly seen in the data and reported by the media. These large changes in numbers are presumably not a function of large fluctuations in new cases or deaths, but instead an artifact of the data reporting process (see *Data reporting process* below and S1 Fig). This process may create unintentional panic or claim to communities.

## Data reporting process

Generally, daily COVID-19 counts are reported from a given data source (e.g., county public health websites) and then extracted by aggregators [36]. The challenge with daily reporting counts is they depend on many varying factors; for example, the time and date these numbers are reported can be nearly immediate or significantly lag. This can be seen in Table 2, where the numbers of new cases often differ depending on the day and time these numbers were reported. Each aggregator reports or publishes the same day counts (e.g., cases and deaths recorded on September 12th 2020), at different times and on different days (e.g., one

**Table 2. Number of new COVID-19 cases by date and aggregator.**

| Date | Aggregator 1 | Aggregator 2 | Aggregator 3 |
|---|---|---|---|
| April 23, 2020 (Thursday) | 87 | 46 | 41 |
| April 24, 2020 (Friday) | 167 | 41 | 0 |
| April 25, 2020 (Saturday) | 0 | 28 | 64 |
| April 26, 2020 (Sunday) | 0 | 36 | 0 |
| April 27, 2020 (Monday) | 0 | 23 | 44 |
| April 28, 2020 (Tuesday) | -59 | 21 | 0 |
| April 29, 2020 (Wednesday) | 32 | 32 | 32 |
| April 30, 2020 (Thursday) | 67 | 19 | 67 |
| May 1, 2020 (Friday) | 0 | 48 | 0 |
| *Mean* | 32.67 | 32.67 | 27.56 |

[a] USAF = USA Facts; NYT = New York Times; JHU = John Hopkins University; BS = BroadStreet; CTP = The COVID Tracking Project

aggregator reports September 12, 2020 counts at the end of day on September 12, 2020, whereas another aggregator may report September 12, 2020 counts on at 8 a.m. on September 13, 2020).

For example in Table 2, both Aggregator 1 and Aggregator 3 report 67 cases on April 30, 2020, whereas Aggregator 2 reported only 19 on April 30, 2020 and the additional 48 cases the following day (May 1, 2020). While the average number of cases are comparable across data aggregators, the number of cases are rarely reliable. Because of daily inconsistencies such as this, reliability estimates were computed and compared using the raw data and a three day moving average. Given the higher reliability of the moving average, this suggests that media and government reports, along with researchers, should consider using a moving average to better represent true trends in COVID-19 cases and deaths given the increase in reliability. This is especially important in early disease tracking when case count sample sizes are small.

### Statistical analyses

To assess the inter-rater reliability (IRR) of COVID-19 aggregators based on new cases and death counts across the counties, states, and United States, a Kappa variant called linearly weighted Cohen's Kappa (LWCK) was used to examine agreement between paired aggregators due to the discrete nature of the data [37, 38]. LWCK inherently takes into account the influence of chance agreement, thus improving the model's sensitivity towards disagreement among rater observation pairs.

After computing LWCK statistics at the county, state, and national levels, choropleth maps were generated for at each level to help visualize COVID-19 spatial event density and understand changes in reliability across aggregators and locations. While several standards have been proposed for IRR, this study employed the standard proposed by Cicchetti and Sparrow [39]: Excellent (0.75 to 1.00), Good (0.60 to 0.75), Fair (0.40 to 0.60), and Poor (0 to 0.40).

### Bayesian

As discussed below, because it is important to examine how reliability results may manifest themselves in commonly used disease tracking statistics, we estimated case fatality rate over time (March 15, 2020 to June 15, 2020) using a novel empirical Bayes approach. Utilizing case fatality rates (number of deaths over a specified period of time) allowed us to track reporting method changes and how this affected the reliability when compared between aggregators. Thus, for each aggregator, we used an empirical Bayes procedure to compute a posterior Beta distribution for each state's case fatality rate (March 15, 2020 to June 15, 2020), based on the number of cases and deaths reported by the corresponding aggregator.

We start with the assumption that the death counts in each state are independently sampled from beta-binomial distributions with common shape parameters $\alpha$ and $\beta$, and with the state-specific number of reported cases. We then estimated the global (U.S. wide) $\alpha$ and $\beta$ parameters via maximum likelihood estimation. These shape parameters form the empirical prior for the subsequent analysis, which from this point forward is a simple Bayesian estimation of binomial proportion. With this common prior, we separately computed the posterior distribution for case fatality rate for each state, based on that state's case and death counts.

## Results

### U.S. level

Table 3 results provide a LWCK reliability comparison across all the aggregators at the U.S. level. These results suggest that mean ($M$) reliability for JHU ($M_{\text{Cases}} = 0.89$, $M_{\text{Death}} = 0.69$),

**Table 3. Reliability comparisons for each pair of aggregators for cases and deaths at U.S. level.**

| Aggregator at U.S. level | Cases | Deaths |
|---|---|---|
| USAF vs. NYT | 0.956 | 0.778 |
| USAF vs. JHU | 0.932 | 0.688 |
| USAF vs. CTP | 0.917 | 0.712 |
| NYT vs. CTP | 0.914 | 0.677 |
| JHU vs. NYT | 0.930 | 0.736 |
| JHU vs. CTP | 0.924 | 0.638 |
| BS vs. USAF | 0.741 | 0.547 |
| BS vs. NYT | 0.745 | 0.583 |
| BS vs. JHU | 0.771 | 0.683 |
| BS vs. CTP | 0.756 | 0.545 |

[a] USAF = USA Facts; NYT = New York Times; JHU = John Hopkins University; BS = BroadStreet; CTP = The COVID Tracking Project

NYT ($M_{Cases}$ = 0.89, $M_{Death}$ = 0.69), USAF ($M_{Cases}$ = 0.89, $M_{Death}$ = 0.68), and CTP ($M_{Cases}$ = 0.88, $M_{Death}$ = 0.64) displayed the highest average inter-rater agreement among the five reported aggregators for both new cases and deaths. On average, BS ($M_{Cases}$ = 0.75, $M_{Death}$ = 0.59) yielded consistently lower inter-rater reliability when compared to other aggregators for both the number of cases and deaths. Interestingly, lower levels of inter-rater agreement were observed across aggregators associated with the number of deaths.

## State level

USAF, NYT, and JHU yielded the highest inter-rater LWCK reliability across all states for both the number of cases and deaths when examining each pair (see Fig 1) and for the average of these aggregator pairs (see Table 3). Intra-rater reliability between aggregators in the form of LWCK state daily counts can be seen in Fig 1, where higher agreement is represented by the darkest green (closest to 1.00) and less agreement is portrayed by the lightest green (0.00). Average reliability for cases and deaths across all aggregators was 0.86 and 0.66, respectively.

Taking a deeper look into the data, high (defined here as kappa ≥ 0.90) mean inter-rater reliability averaged across aggregator pairings (see Table 4) was observed across all aggregator state case comparisons for LA, VA, SD, AZ, CT, MD, NJ, and FL when examining the number of cases. Further, the average reliability for deaths was high for only SD, ME, OK, and CT. Note, several states (i.e., RI, OK, MI, NV, and KS) had unacceptable average reliability statistics (defined here as kappa ≤ 0.70) associated with the number of cases, and reliability was even worse for the death rates of 29 states (see Table 4). Comparing states that have high versus low mean inter-rater reliability averaged across aggregator pairings is important to note for better overall understanding and for the replicating of high mean inter-rater reliability reporting practices.

Based on these results and other results provided in Table 4, the number of cases data were consistently more reliable than the number of deaths. Further depending on the aggregator, the variance and range in reliability within a state can be large, regardless of whether it is the number of cases or deaths. Although several examples exist, for the average number of cases and death data, CTP had an average reliability of 0.76 within NY whereas BS only had a reliability of 0.23 within NY. This is important to note as it suggests that reliability is a function of both the aggregator and state.

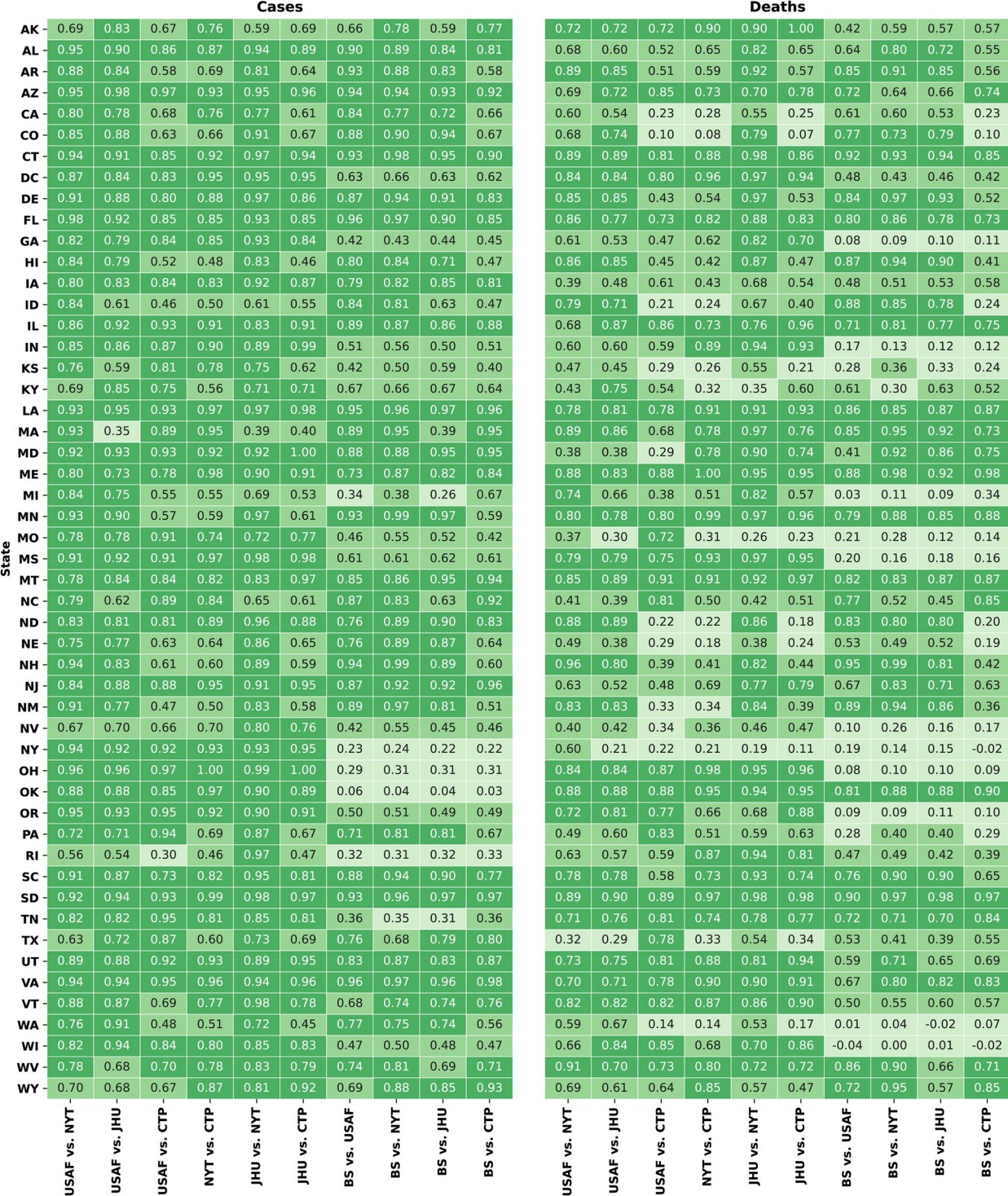

**Fig 1. Provides the intra-rater reliability between each aggregator pair as Kappas state daily counts.** Darker (1.00) to lighter (0.00) color indicates more to less agreement, respectively.

Table 4. Average kappa (Fig 3) for each aggregator by state.

| State | Cases | | | | | | Deaths | | | | | |
|---|---|---|---|---|---|---|---|---|---|---|---|---|
| | USAF | NYT | JHU | BS | CTP | *M* | USAF | NYT | JHU | BS | CTP | *M* |
| AK | 0.71 | 0.71 | 0.67 | 0.70 | 0.72 | 0.70 | 0.64 | 0.78 | 0.80 | 0.54 | 0.80 | 0.71 |
| AL | 0.90 | 0.91 | 0.89 | 0.86 | 0.86 | 0.88 | 0.61 | 0.74 | 0.70 | 0.68 | 0.59 | 0.66 |
| AR | 0.81 | 0.81 | 0.78 | 0.80 | 0.62 | 0.76 | 0.77 | 0.83 | 0.80 | 0.79 | 0.56 | 0.75 |
| AZ | 0.96 | 0.94 | 0.95 | 0.93 | 0.94 | 0.95 | 0.74 | 0.69 | 0.72 | 0.69 | 0.77 | 0.72 |
| CA | 0.77 | 0.78 | 0.72 | 0.75 | 0.68 | 0.74 | 0.49 | 0.50 | 0.47 | 0.49 | 0.25 | 0.44 |
| CO | 0.81 | 0.83 | 0.85 | 0.85 | 0.66 | 0.80 | 0.57 | 0.57 | 0.60 | 0.60 | 0.09 | 0.48 |
| CT | 0.91 | 0.95 | 0.94 | 0.94 | 0.90 | 0.93 | 0.88 | 0.92 | 0.92 | 0.91 | 0.85 | 0.90 |
| DC | 0.79 | 0.86 | 0.84 | 0.63 | 0.84 | 0.79 | 0.74 | 0.80 | 0.80 | 0.45 | 0.78 | 0.71 |
| DE | 0.87 | 0.93 | 0.90 | 0.89 | 0.84 | 0.89 | 0.74 | 0.83 | 0.82 | 0.81 | 0.51 | 0.74 |
| FL | 0.93 | 0.93 | 0.90 | 0.92 | 0.85 | 0.91 | 0.79 | 0.86 | 0.82 | 0.79 | 0.78 | 0.81 |
| GA | 0.72 | 0.76 | 0.75 | 0.43 | 0.75 | 0.68 | 0.42 | 0.53 | 0.54 | 0.09 | 0.47 | 0.41 |
| HI | 0.74 | 0.75 | 0.69 | 0.71 | 0.48 | 0.67 | 0.76 | 0.78 | 0.77 | 0.78 | 0.44 | 0.71 |
| IA | 0.82 | 0.84 | 0.87 | 0.82 | 0.84 | 0.84 | 0.49 | 0.50 | 0.56 | 0.52 | 0.54 | 0.52 |
| ID | 0.69 | 0.69 | 0.60 | 0.69 | 0.49 | 0.63 | 0.65 | 0.64 | 0.64 | 0.69 | 0.27 | 0.58 |
| IL | 0.90 | 0.87 | 0.88 | 0.88 | 0.91 | 0.89 | 0.78 | 0.75 | 0.84 | 0.76 | 0.83 | 0.79 |
| IN | 0.77 | 0.80 | 0.81 | 0.52 | 0.82 | 0.75 | 0.49 | 0.64 | 0.65 | 0.14 | 0.64 | 0.51 |
| KS | 0.64 | 0.70 | 0.64 | 0.47 | 0.65 | 0.62 | 0.37 | 0.41 | 0.39 | 0.30 | 0.25 | 0.34 |
| KY | 0.74 | 0.66 | 0.74 | 0.66 | 0.66 | 0.69 | 0.58 | 0.35 | 0.58 | 0.51 | 0.49 | 0.50 |
| LA | 0.94 | 0.96 | 0.97 | 0.96 | 0.96 | 0.96 | 0.81 | 0.86 | 0.88 | 0.86 | 0.87 | 0.86 |
| MA | 0.76 | 0.80 | 0.38 | 0.80 | 0.80 | 0.71 | 0.82 | 0.90 | 0.88 | 0.87 | 0.74 | 0.84 |
| MD | 0.91 | 0.91 | 0.95 | 0.91 | 0.95 | 0.93 | 0.36 | 0.74 | 0.72 | 0.74 | 0.64 | 0.64 |
| ME | 0.76 | 0.89 | 0.84 | 0.81 | 0.88 | 0.84 | 0.87 | 0.95 | 0.91 | 0.94 | 0.95 | 0.92 |
| MI | 0.62 | 0.62 | 0.56 | 0.41 | 0.58 | 0.56 | 0.45 | 0.55 | 0.53 | 0.14 | 0.45 | 0.42 |
| MN | 0.83 | 0.87 | 0.86 | 0.87 | 0.59 | 0.80 | 0.79 | 0.91 | 0.89 | 0.85 | 0.91 | 0.87 |
| MO | 0.73 | 0.70 | 0.70 | 0.49 | 0.71 | 0.67 | 0.40 | 0.30 | 0.23 | 0.19 | 0.35 | 0.30 |
| MS | 0.84 | 0.87 | 0.88 | 0.61 | 0.87 | 0.81 | 0.63 | 0.71 | 0.72 | 0.18 | 0.70 | 0.59 |
| MT | 0.83 | 0.82 | 0.90 | 0.90 | 0.89 | 0.87 | 0.87 | 0.88 | 0.91 | 0.85 | 0.91 | 0.88 |
| NC | 0.79 | 0.77 | 0.63 | 0.81 | 0.82 | 0.76 | 0.60 | 0.46 | 0.45 | 0.65 | 0.67 | 0.56 |
| ND | 0.80 | 0.89 | 0.89 | 0.84 | 0.85 | 0.86 | 0.71 | 0.69 | 0.68 | 0.66 | 0.21 | 0.59 |
| NE | 0.73 | 0.79 | 0.79 | 0.79 | 0.64 | 0.75 | 0.42 | 0.38 | 0.38 | 0.43 | 0.22 | 0.37 |
| NH | 0.83 | 0.86 | 0.80 | 0.85 | 0.60 | 0.79 | 0.77 | 0.79 | 0.71 | 0.79 | 0.41 | 0.70 |
| NJ | 0.87 | 0.90 | 0.91 | 0.91 | 0.93 | 0.91 | 0.58 | 0.73 | 0.70 | 0.71 | 0.65 | 0.67 |
| NM | 0.76 | 0.80 | 0.75 | 0.80 | 0.52 | 0.72 | 0.72 | 0.74 | 0.73 | 0.76 | 0.36 | 0.66 |
| NV | 0.61 | 0.68 | 0.68 | 0.47 | 0.65 | 0.62 | 0.32 | 0.37 | 0.38 | 0.17 | 0.33 | 0.31 |
| NY | 0.75 | 0.76 | 0.75 | 0.23 | 0.76 | 0.65 | 0.30 | 0.29 | 0.17 | 0.12 | 0.13 | 0.20 |
| OH | 0.80 | 0.82 | 0.82 | 0.31 | 0.82 | 0.71 | 0.66 | 0.72 | 0.71 | 0.09 | 0.72 | 0.58 |
| OK | 0.67 | 0.70 | 0.68 | 0.04 | 0.69 | 0.55 | 0.86 | 0.91 | 0.91 | 0.87 | 0.92 | 0.90 |
| OR | 0.83 | 0.82 | 0.81 | 0.50 | 0.82 | 0.75 | 0.60 | 0.54 | 0.62 | 0.10 | 0.60 | 0.49 |
| PA | 0.77 | 0.77 | 0.76 | 0.75 | 0.74 | 0.76 | 0.55 | 0.50 | 0.55 | 0.34 | 0.56 | 0.50 |
| RI | 0.43 | 0.58 | 0.57 | 0.32 | 0.39 | 0.46 | 0.57 | 0.73 | 0.68 | 0.44 | 0.66 | 0.62 |
| SC | 0.85 | 0.91 | 0.88 | 0.87 | 0.78 | 0.86 | 0.72 | 0.84 | 0.84 | 0.80 | 0.68 | 0.78 |
| SD | 0.93 | 0.96 | 0.97 | 0.96 | 0.96 | 0.96 | 0.89 | 0.95 | 0.96 | 0.95 | 0.95 | 0.94 |
| TN | 0.74 | 0.71 | 0.70 | 0.35 | 0.73 | 0.65 | 0.75 | 0.73 | 0.75 | 0.74 | 0.79 | 0.75 |
| TX | 0.74 | 0.66 | 0.73 | 0.76 | 0.74 | 0.73 | 0.48 | 0.40 | 0.39 | 0.47 | 0.50 | 0.45 |
| UT | 0.88 | 0.90 | 0.89 | 0.85 | 0.92 | 0.89 | 0.72 | 0.78 | 0.79 | 0.66 | 0.83 | 0.76 |
| VA | 0.95 | 0.95 | 0.95 | 0.97 | 0.96 | 0.96 | 0.71 | 0.83 | 0.83 | 0.78 | 0.85 | 0.80 |

*(Continued)*

**Table 4.** (Continued)

| State | Cases | | | | | | Deaths | | | | | |
|---|---|---|---|---|---|---|---|---|---|---|---|---|
| | USAF | NYT | JHU | BS | CTP | *M* | USAF | NYT | JHU | BS | CTP | *M* |
| VT | 0.78 | 0.84 | 0.84 | 0.73 | 0.75 | 0.79 | 0.74 | 0.78 | 0.80 | 0.55 | 0.79 | 0.73 |
| WA | 0.73 | 0.68 | 0.70 | 0.70 | 0.50 | 0.66 | 0.35 | 0.33 | 0.34 | 0.02 | 0.13 | 0.23 |
| WI | 0.77 | 0.74 | 0.78 | 0.48 | 0.73 | 0.70 | 0.58 | 0.51 | 0.60 | -0.01* | 0.59 | 0.45 |
| WV | 0.72 | 0.80 | 0.75 | 0.74 | 0.74 | 0.75 | 0.80 | 0.83 | 0.70 | 0.78 | 0.74 | 0.77 |
| WY | 0.68 | 0.81 | 0.81 | 0.84 | 0.85 | 0.80 | 0.67 | 0.77 | 0.56 | 0.77 | 0.71 | 0.69 |

[a] USAF = USA Facts; NYT = New York Times; JHU = John Hopkins University; BS = BroadStreet; CTP = The COVID Tracking Project.

*Kappa can produce negative values 0 is random agreement among raters; 1 is complete agreement; less than 0 is generally interpreted as "no agreement." [40]

## County level

The maps in Figs 2 and 3 provide reliability statistics at the county level for each aggregator pair based on the number of cases and deaths, respectively. A higher rate of agreement in both figures is represented by dark purple (1.00) whereas a lower rate of agreement is represented by lime green (0.00) (see Figs 2 and 3). The presented results suggest that county-level reliability appears clustered within states, rather than scattered throughout the country. Perhaps more interesting is that certain pairs of aggregators are more reliable in some counties than others, thus pointing to the concern that the data collection methods used may result in significantly different conclusions (both for political and research purposes) at local levels. While it is clear from Table 4 that the level of reliability is often state dependent, Figs 2 and 3 demonstrate there is also significant variation within states.

**Bayesian.** It is essential to examine case fatality rate (CFR) estimates and not just reliability statistics to explore how reliability estimates may translate into actual statistics used to track

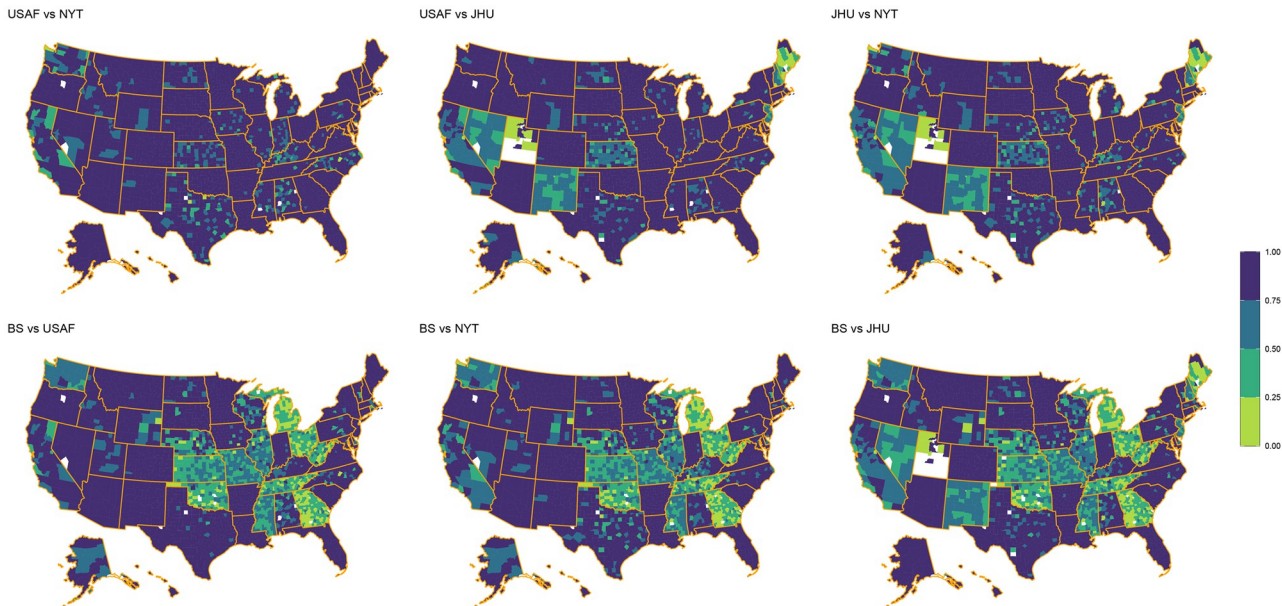

**Fig 2. Provides the reliability within each county for the number of COVID-19 <u>cases</u> using the reliability categories proposed by Cicchetti and Sparrow [39].** Darker (1.00) to lighter (0.00) color indicates more to less agreement. For kappa, smaller sample size differences won't penalize smaller value differences.

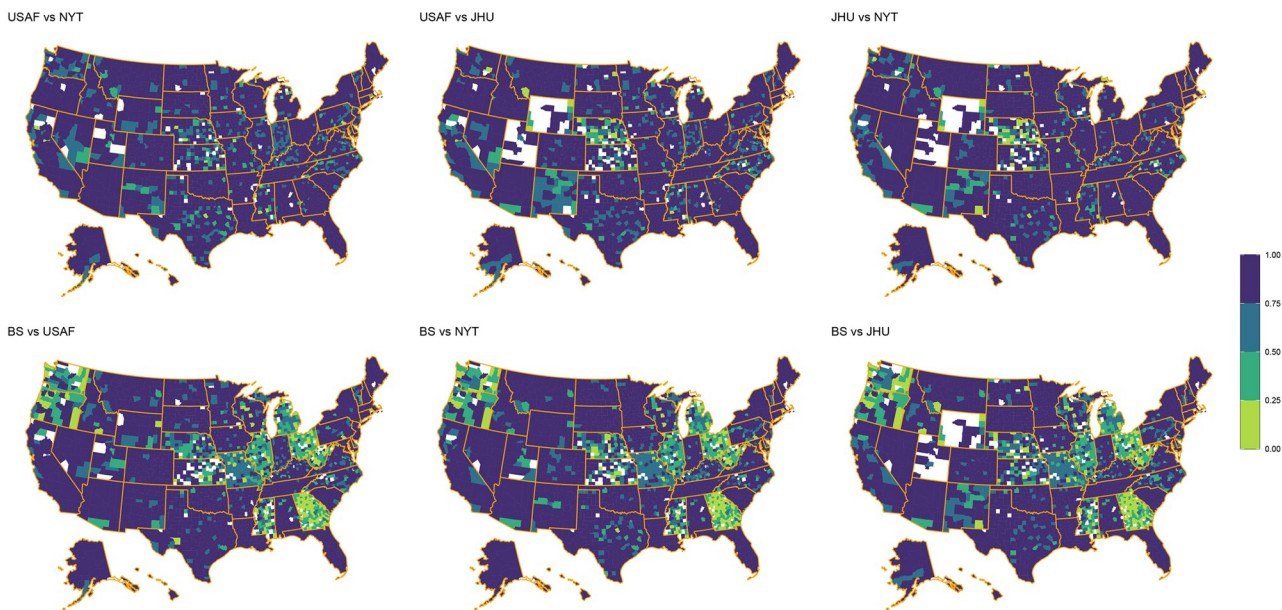

**Fig 3. Provides the reliability within each county for the number of COVID-19 <u>deaths</u> using the reliability categories proposed by Cicchetti and Sparrow.** Darker (1.00) to lighter (0.00) color indicates more to less agreement.

COVID-19. Since the empirical Bayes approach was repeated over time, we can see how the estimate for case fatality rate develops as more data becomes available (Fig 4). Fig 4 depicts the progression of case fatality rates on March 15, 2020, April 15, 2020, May 15, 2020, and June 15, 2020. As time passes and more cases and deaths are reported, the intervals (represented by colored horizontal bars in Figs 4 and 5) narrow, giving a more precise estimate for the case fatality rate. In addition, for most states, the estimates corresponding to the different aggregators begin to align with each other over time.

If we look a bit closer at the data from June 30, 2020 in Fig 5 we can see how low reliability manifests itself in varying fatality estimates. For example, in NY we see reliability estimates ranging from 0.12 to 0.30 (see Table 4), which results in an aggregate fatality range of 0.063 to 0.082 (26.20% difference). If we go back and look at CFR estimate differences across aggregators for NY we see: 0.008 to 0.016 (119.21% difference) on March 15, 2020; 0.050 to 0.076 (34.28% difference) on April 15, 2020; 0.065 to 0.083 (25.03% difference) on May 15, 2020; and 0.064 to 0.082 to (24.68% difference) on June 15, 2020. This also means our confidence in estimates will be low early on, especially when sample sizes are small. Looking at AK credible intervals (CI), its smallest lower CI is 0.007 and its largest upper CI is 0.024, which are fairly large differences for potential point estimates of fatality rates. What this implies is that low reliability, assessed by estimating the inter-rater reliability between aggregators, can lead to significant differences across aggregators in calculated disease tracking statistics, particularly early in a pandemic.

## Discussion

This study compared the reliability of COVID-19 death and cases count data across national, state, and county-levels between data aggregators. As expected, given the larger sample sizes, reliability for both cases and deaths was higher at the national level across aggregators than at state and county levels. However, death count reliability was typically lower than reliability for reported cases. Variation in reliability remained across aggregators and suggests that

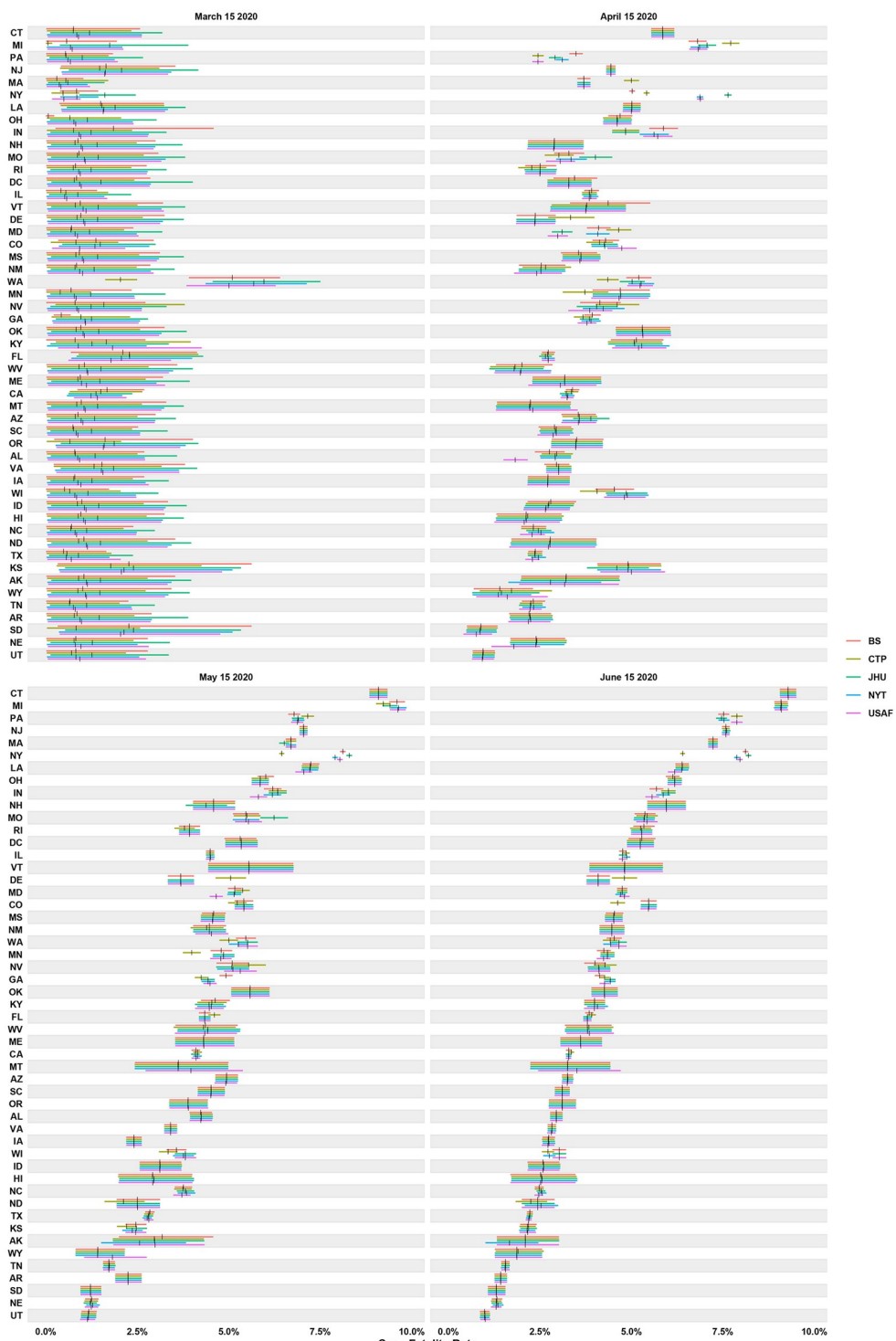

**Fig 4. Time-progression of case fatality rate across states and sources.** The horizontal bars represent the 90% equal-tail credible intervals, with the vertical black bars indicating the posterior means.

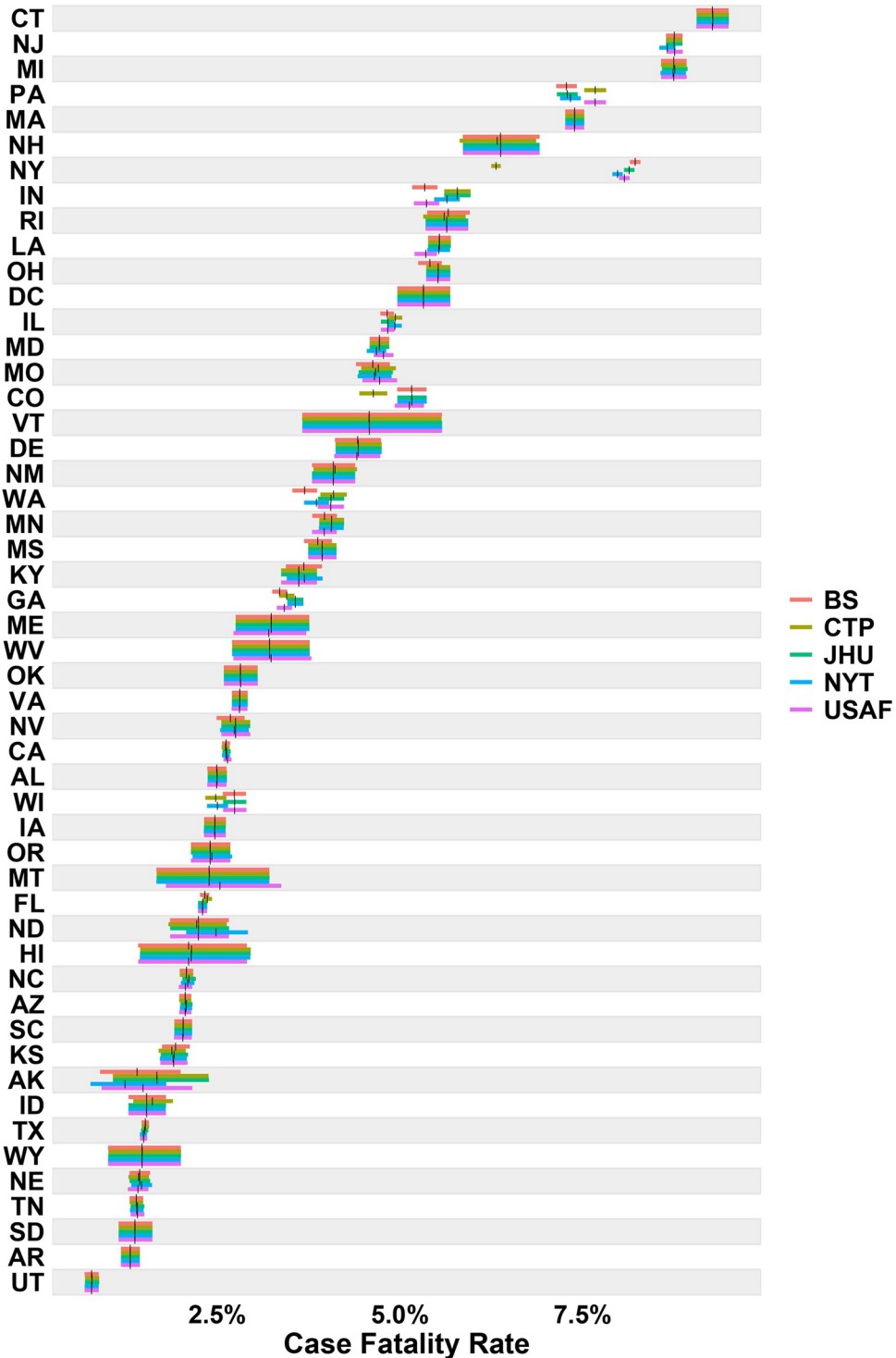

**Fig 5. Aggregated (June 30, 2020) case fatality rate across states and sources.** The horizontal bars represent the 90% equal-tail credible intervals, with the vertical black bars indicating the posterior means.

aggregator choice could have a significant impact on any data analysis or subsequent action based on the data.

These differences can partially be explained by the intended purposes and collections methods of each aggregator, making it ever more important that aggregators clearly define data collection methods and define terms used (i.e., cases and deaths). USAF [27], JHU [28], and NYT [29] have been reporting daily cases early in the pandemic and attempt to publish case and death totals in near "real-time" or as they are reported. They occasionally use county-level health departments as a source in instances where the state health department is lagging significantly behind them. Conversely, CTP [30] exclusively uses state health departments as a source to ensure their data are consistent. Broadstreet has emphasized updating data to ensure it is following a logical trend and, when possible, updating historic data to reflect more epidemiologically significant dates, such as date of death or date of symptom onset. These disparate approaches are important for different reasons (e.g., historical accuracy for retrospective analysis) and likely caused some differences in inter-rater reliability. This level of agreement was assessed by estimating the inter-rater reliability between aggregators.

State level reliability (Fig 1 and Table 4) demonstrated notable findings as well. First, reliability is not equal across states, suggesting that individual state practices and policies considerably influence the data's reliability. Second, some reliability estimates are extremely poor suggesting analysis of these data could produce inconsistent findings and biased results/inferences. These results imply the need for standardization of collection and reporting methods across states, which would increase both reliability and validity of the data. Ideally, this could also be done on a national level, enhancing the data's reliability and validity while also guaranteeing that the data, along with the cohesive reporting methods, is made publicly available.

USAF, JHU, NYT, CPT, and BS use various data collection and quality assurance methods, as well as sources [26–30]. Using state public health departments as a data source requires less maintenance and is more sustainable, but county health departments tend to be more up-to-date than state health departments, which may be of paramount importance early in a pandemic. County health departments also report data based on Council of State and Territorial Epidemiologists (CSTE) guidelines and case definitions, which can help avoid including unreasonable cases in data [41]. However, this may potentially cause datasets to include duplicate cases and cases with an unreasonable definition, making the sum of all counties overestimate the national total.

While data scraping is a quick and accurate way to enter data, it is fraught with technical challenges (e.g., updating web sites breaks the scrapping code) and in our experience is not yet feasible, quicker, or any more accurate than manual crowdsourcing data entry. Moreover, data scraping requires human eyes monitoring it, which may cause data to be missed in instances where the health department alters their website. Likewise, a pipeline may fail to fetch data if the health department changes how their website is formatted; especially a potential issue if volunteers are not expecting this particular county to update daily, and may not notice a fetching issue. Updating historical data to include cases and deaths by date of symptom onset or death provides significant information when analyzing spread of virus and the effectiveness of preventative measures.

In regards to case and death data, the NYT attributes cases to the location they are being treated, which may provide a more accurate picture of how the virus is spreading in particular counties and states compared to using solely the date reported. Despite this, the information they use to assign cases to counties is inconsistently provided, and the data may include out of state visitors in state totals; if these cases are also reported back to the home state, these cases will be counted twice on the national level.

## Limitations

Consequently, as a result of challenges posed in the many stages of data analyses, the reliability and validity of these statistics is critical when creating policies to protect the public and accurately modeling the disease. Disease data validity is imperative and should be the primary objective for any institute, as without validity there can be no reliability. Given that validity cannot be assessed without significant agency and/or government oversight, this study sought to evaluate COVID-19 data reliability, providing insight into the consistency of data across different sources. Due to this, the first limitation of this study is the lack of validity being addressed.

Fundamentally, the validity of any statistical analysis is based on the quality of data collected [39, 41–46]. Moreover, it is critical that aggregators are transparent in their data collection process so users can judge the validity of their process and can understand discrepancies in numbers across data collection sources. An important caveat is the validity of the final data source is largely dependent on the initial sources providing the data (e.g., state officials and hospitals). For this reason, it is critical that mechanisms are also put in place to evaluate the reliability and validity of data sources at this level. Unfortunately, to our knowledge there is currently no mechanism in place to evaluate this process or the accuracy of the data collected [2], resulting in uncertainty regarding the exact reasoning for data discrepancies across certain states and counties.

Due to validity concerns, several findings were clear when evaluating the reliability of reported daily cases and deaths across aggregators. First, a 3-day moving average is likely needed to ensure reliability across aggregators and eliminate large spikes or dips in the data associated with validity issues. To account for this, Cohen's Kappa was used, though with limitations, such as the potential for paradoxical coefficients such that high agreement yield zero-value coefficients, negative coefficients, or abnormally low coefficients [47, 48]. Modifying the moving average counts by +1 did improve overall kappa performance, however a handful of paradoxical results still occurred. Moreover, reliability was not examined for changes over time and the data in this study only extends to June 15, 2020. Reporting methods may have since updated or changed since this study occurred, which may result in some inaccuracies.

While it would be ideal if cases were reported using the date of infection or death rather than when the event was reported and all aggregators used these dates, this was not the case and frequently resulted in significant spikes on certain days (e.g., cases often dropped over the weekend and spiked on Monday or Tuesday, with the level of these spikes often being agency or county dependent). While the aforementioned example associated with data spikes certainly impacts the data's validity, one should be cognizant that it should not impact the reliability (i.e., aggregators should reliability report those spikes). With that said, it is clear from our evaluation of the aggregator's data that the practices applied across aggregators is not consistent (Table 3), thus practices should be put in place to increase reliability rather than relying on data smoothing methods to reduce the impact of inconsistent reporting.

## Conclusions and relevance

The primary conclusion from this study is that the United States needs a national public data reporting system that is free from the inconsistencies and data discrepancies that result from decentralized data collection and aggregation. The technology to make this happen currently exists. More than 95% of U.S. hospitals use an Electronic Health Record system [5], which can be integrated into a near-real-time data reporting infrastructure to share data between local, state, and national public health agencies. Additionally, the CDC maintains a National Notifiable Diseases Surveillance System (NNDSS), which is used to aggregate data on nationally

reportable and notifiable diseases. COVID-19 data are submitted electronically to the CDC by state or jurisdictional health departments via the COVID-19 Electronic Laboratory Reporting system. However, participation in reporting to the NNDSS varies widely between states because participation in the program is entirely voluntary. The problem, therefore, lies in the initial collection and eventual reporting of data from the states. Differences in the underlying Infection Fatality Rate (deaths per true number of infections, rather than deaths per detected/reported cases) would cause discrepancies if certain states had a more vulnerable populace than others, due to demographics such as age or socioeconomic status.

Another likely contributor is differences in reporting. The lack of a nation-wide standard for reporting deaths means that different states may be more or less stringent in attributing deaths to the virus. A third possible source for disagreement across states is discrimination in testing. Due to limited availability of testing, some states became more restrictive in providing free tests to the public. Tests in such states were prioritized towards those exhibiting more severe symptoms, and consequently could have introduced case sampling bias towards a higher-risk subset of the greater population of infected individuals. Through future research, different databases and public sources will be incredibly valuable in the tracking and documentation of cases and deaths [49, 50].

Standardizing infectious disease data collection and dissemination would empower practitioners to do more linking to other variables and analysis. For example, the Area Deprivation Index is a powerful indicator of many health outcomes [50–52]. The Centers for Disease Control and Prevention (CDC) reports social inequality and health systems issues as a cause for an increased risk of health and socioeconomic impacts as a result of COVID-19 for these groups [53, 54]. Data reporting for race began in early April, with Louisiana being the first to report data [54–56]. Immediately, disparities in mortality deaths were noticed, and a June 2020 report by the CDC confirmed this disparity was widespread. Ultimately, the United States needs to nationally mandate explicit methods for reportable infectious diseases. This is a political problem, spanning policy, communication, and public health sectors. Public health funding should be directed toward the development of a national reporting database that clearly identifies COVID-19 cases and fatalities, as well as consistent reporting procedures, effectively modernizing disease reporting in the US.

## Supporting information

**S1 File.**
(DOCX)

**S1 Fig. Visualization of Washington state's COVID-19 protocol.** Note: Time period of this chart is July 10, 2020. Process may have been updated.
(DOCX)

## Acknowledgments

We'd like to give a special thank you to everyone who participated in the COVID-19 Data Project. We'd also like to thank all the universities that have helped us along the way and allowed us to borrow their wonderful students for a good cause: Simmons University, New York University, Temple University, George Washington University, and many others!

## Author Contributions

**Conceptualization:** Daniel A. Sass, Thomas A. Schmitt.

**Data curation:** April R. Miller, Samin Charepoo, Erik Yan, Zachary J. Sturgeon, Grace Gibbon, Patrick N. Balius, Cedonia S. Thomas, James B. Walters, Thomas A. Schmitt.

**Formal analysis:** Erik Yan, Ryan W. Frost, Daniel A. Sass, Thomas A. Schmitt.

**Funding acquisition:** Tracy L. Flood, Thomas A. Schmitt.

**Investigation:** April R. Miller, Samin Charepoo, Thomas A. Schmitt.

**Methodology:** Erik Yan, Ryan W. Frost, Daniel A. Sass, Thomas A. Schmitt.

**Project administration:** Thomas A. Schmitt.

**Resources:** Thomas A. Schmitt.

**Supervision:** Thomas A. Schmitt.

**Validation:** Thomas A. Schmitt.

**Visualization:** Thomas A. Schmitt.

**Writing – original draft:** April R. Miller, Samin Charepoo, Erik Yan, Zachary J. Sturgeon, Cedonia S. Thomas, Melanie A. Schmitt, Daniel A. Sass, James B. Walters, Thomas A. Schmitt.

**Writing – review & editing:** April R. Miller, Samin Charepoo, Erik Yan, Ryan W. Frost, Zachary J. Sturgeon, Grace Gibbon, Patrick N. Balius, Daniel A. Sass, James B. Walters, Tracy L. Flood, Thomas A. Schmitt.

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
