## [Decision Letter · Decision Letter 0]

15 Jul 2021

PONE-D-21-19745

Reliability of COVID-19 data: An evaluation and reflection

PLOS ONE

Dear Dr. Schmitt,

Thank you for submitting your manuscript to PLOS ONE. After careful consideration, we feel that it has merit but does not fully meet PLOS ONE’s publication criteria as it currently stands. Therefore, we invite you to submit a revised version of the manuscript that addresses the points raised during the review process.

Please address the issues and revise accordingly.

We look forward to receiving your revised manuscript.

Kind regards,

Academic Editor

PLOS ONE

Journal Requirements:

We note that one or more of the authors have an affiliation to the commercial funders of this research study : BroadStreet Health.

2.1. Please provide an amended Funding Statement declaring this commercial affiliation, as well as a statement regarding the Role of Funders in your study. If the funding organization did not play a role in the study design, data collection and analysis, decision to publish, or preparation of the manuscript and only provided financial support in the form of authors' salaries and/or research materials, please review your statements relating to the author contributions, and ensure you have specifically and accurately indicated the role(s) that these authors had in your study. You can update author roles in the Author Contributions section of the online submission form.

2.2. Please also provide an updated Competing Interests Statement declaring this commercial affiliation along with any other relevant declarations relating to employment, consultancy, patents, products in development, or marketed products, etc.  

"Thank you to BroadStreet and Franciscan Health for continued support."

Additionally, because some of your funding information pertains to [commercial funding//patents], we ask you to provide an updated Competing Interests statement, declaring all sources of commercial funding.

In your Competing Interests statement, please confirm that your commercial funding does not alter your adherence to PLOS ONE Editorial policies and criteria by including the following statement: "This does not alter our adherence to PLOS ONE policies on sharing data and materials.” as detailed online in our guide for authors  http://journals.plos.org/plosone/s/competing-interests.  If this statement is not true and your adherence to PLOS policies on sharing data and materials is altered, please explain how.

Please include the updated Competing Interests Statement and Funding Statement in your cover letter. We will change the online submission form on your behalf.

4. One of the noted authors is a group or consortium [COVID-19 Data Project]. In addition to naming the author group, please list the individual authors and affiliations within this group in the acknowledgments section of your manuscript. Please also indicate clearly a lead author for this group along with a contact email address.

5. We note that Figures 2 and 3 in your submission contain map images which may be copyrighted. All PLOS content is published under the Creative Commons Attribution License (CC BY 4.0), which means that the manuscript, images, and Supporting Information files will be freely available online, and any third party is permitted to access, download, copy, distribute, and use these materials in any way, even commercially, with proper attribution. For these reasons, we cannot publish previously copyrighted maps or satellite images created using proprietary data, such as Google software (Google Maps, Street View, and Earth). For more information, see our copyright guidelines: http://journals.plos.org/plosone/s/licenses-and-copyright.

5.1.    You may seek permission from the original copyright holder of Figures 2 and 3 to publish the content specifically under the CC BY 4.0 license. 

5.2.    If you are unable to obtain permission from the original copyright holder to publish these figures under the CC BY 4.0 license or if the copyright holder’s requirements are incompatible with the CC BY 4.0 license, please either i) remove the figure or ii) supply a replacement figure that complies with the CC BY 4.0 license. Please check copyright information on all replacement figures and update the figure caption with source information. If applicable, please specify in the figure caption text when a figure is similar but not identical to the original image and is therefore for illustrative purposes only.

Reviewers' comments:

Reviewer's Responses to Questions

**Comments to the Author**

1. Is the manuscript technically sound, and do the data support the conclusions?

Reviewer #1: Partly

Reviewer #2: Yes

Reviewer #3: Yes

Reviewer #4: No

2. Has the statistical analysis been performed appropriately and rigorously? 

Reviewer #1: Yes

Reviewer #2: Yes

Reviewer #3: Yes

Reviewer #4: Yes

3. Have the authors made all data underlying the findings in their manuscript fully available?

Reviewer #1: Yes

Reviewer #2: Yes

Reviewer #3: No

Reviewer #4: No

4. Is the manuscript presented in an intelligible fashion and written in standard English?

Reviewer #1: No

Reviewer #2: No

Reviewer #3: Yes

Reviewer #4: No

5. Review Comments to the Author

Reviewer #1: The authors have described the COVID-19 data collection process, examined the reliability of COVID-19 (cases and deaths) data aggregators, and examined the case fatality rate across states and sources.

However, there are some issues that needs to addressed

Major issues

1. The main concern is the readability of the manuscript. Though it has been conducted rigorously, it poses difficulty for the readers to comprehend the results and their implications.

2. Was there any eligibility criteria for selection of volunteers as well as sources included in the study. If Yes; the criteria's can be listed?

3. Were steps undertaken to ensure the transparency of aggregators in data collection process. If yes, the steps undertaken can be stated.

4. 'Result' section need strengthening and succinct description.

5. Generalizability of the findings of this study can be stated.

Minor issues

1. All the figures are cluttered and are difficult to comprehend.

Reviewer #2: This is a paper describing and evaluating the reliability of COVID-19 data from various sources in the United states at the national and county level.

The background and aims are clear and well documented. Introduction could be shortened specifically the first paragraph where description should be "to the point".The methodology is robust, detailed, and sound. The discussion and conclusions are appropriate. The references are up to date.

The paper needs minor writing edits.

Reviewer #3: 1. The paper is well written.

2. Tables and graphics are well presented.

3. Few of the statistical techniques mentioned in statistical analysis like intraclass correlation coefficient with mixed effects, Spearman and Pearson correlation were not discussed in results.

Reviewer #4: Title: Reliability of COVID-19 data: An evaluation and reflection

Introduction:

First paragraph is too complex and does not have a link with information and reliability.

Second and third paragraphs are adequate and reflect the objective of the present paper.

The aim is not clear for me. Authors should specify.

Methods:

“Starting on March 16th, 2020, the Broadstreet team (consisting of approximately

120 volunteers) began tracking diagnosed cumulative cases of, and deaths due to

COVID-19 reported by state and county governments” What is the reference for this information? Website? Please specify.

Methods are too complex. So it is the table 1.

There is a need and better definition of data reliability in the text.

Results and Discussion are too specific. Although the subject is indeed interesting the presentation is very specific and complex. I would suggest a specific journal (Science Computer, for an example).

Finally there is no information regarding ethics approval. Considering that patient information, covid cases and deaths are being recorded, there should be information about the research ethics committee.

6. PLOS authors have the option to publish the peer review history of their article (what does this mean?). If published, this will include your full peer review and any attached files.

Reviewer #1: **Yes: **Mahalaqua Nazli Khatib

Reviewer #2: No

Reviewer #3: No

Reviewer #4: No

---

## [Author Response · Author response to Decision Letter 0]

7 Oct 2021

Thank you for stating the following in the Acknowledgments Section of your manuscript: "Thank you to BroadStreet and Franciscan Health for continued support." 

We note that you have provided funding information that is not currently declared in your Funding Statement. However, funding information should not appear in the Acknowledgments section or other areas of your manuscript. We will only publish funding information present in the Funding Statement section of the online submission form. Please remove any funding-related text from the manuscript and let us know how you would like to update your Funding Statement. Currently, your Funding Statement reads as follows: "The author(s) received no specific funding for this work." Additionally, because some of your funding information pertains to [commercial funding//patents], we ask you to provide an updated Competing Interests statement, declaring all sources of commercial funding.

In your Competing Interests statement, please confirm that your commercial funding does not alter your adherence to PLOS ONE Editorial policies and criteria by including the following statement: "This does not alter our adherence to PLOS ONE policies on sharing data and materials.” as detailed online in our guide for authors http://journals.plos.org/plosone/s/competing-interests. If this statement is not true and your adherence to PLOS policies on sharing data and materials is altered, please explain how.

Please include the updated Competing Interests Statement and Funding Statement in your cover letter. We will change the online submission form on your behalf.

Response: Thank you for these notes. We have now updated the ‘Acknowledgements’ section of the manuscript by removing any funding information. The funding statement and competing interests statement has also been updated in the cover letter.

Thank you for indicating that the map images in Figure 2 & 3 are not original. To ensure that they are compliant with our copyright policies, however, we ask that you provide additional information regarding the following points:

Please indicate where the base maps/shapefiles used to create the map images were retrieved from, and provide a direct link to the source.

Please indicate the source of the map data presented, and provide a direct link to the source, as relevant.

Explain what software was used to create the map.

Update the figure captions in your manuscript file accordingly.

Remove the figure images from the manuscript file and upload them all instead as individual TIFF or PNG figure files, with the "Figure" item type in Editorial Manager.

Response: As previously stated, all figures within the manuscript are original. They were created by the authors using original data in open source R stats.

---

## [Decision Letter · Decision Letter 1]

15 Nov 2021

PONE-D-21-19745R1Reliability of COVID-19 data: An evaluation and reflectionPLOS ONE

Dear Dr. Schmitt,

Thank you for submitting your manuscript to PLOS ONE. After careful consideration, we feel that it has merit but does not fully meet PLOS ONE’s publication criteria as it currently stands. Therefore, we invite you to submit a revised version of the manuscript that addresses the points raised during the review process. Please revise the manuscript to address all the reviewer's comments in a point-by-point response in order to ensure it is meeting the journal's publication criteria. Please note that the revised manuscript will need to undergo further review, we thus cannot at this point anticipate the outcome of the evaluation process.

We look forward to receiving your revised manuscript.

Kind regards,

Miquel Vall-llosera Camps

Senior Editor

PLOS ONE

Journal Requirements:

Reviewers' comments:

Reviewer's Responses to Questions

**Comments to the Author**

1. If the authors have adequately addressed your comments raised in a previous round of review and you feel that this manuscript is now acceptable for publication, you may indicate that here to bypass the “Comments to the Author” section, enter your conflict of interest statement in the “Confidential to Editor” section, and submit your "Accept" recommendation.

Reviewer #1: All comments have been addressed

Reviewer #3: All comments have been addressed

Reviewer #4: (No Response)

2. Is the manuscript technically sound, and do the data support the conclusions?

Reviewer #1: Yes

Reviewer #3: Yes

Reviewer #4: Partly

3. Has the statistical analysis been performed appropriately and rigorously? 

Reviewer #1: Yes

Reviewer #3: Yes

Reviewer #4: Yes

4. Have the authors made all data underlying the findings in their manuscript fully available?

Reviewer #1: Yes

Reviewer #3: Yes

Reviewer #4: Yes

5. Is the manuscript presented in an intelligible fashion and written in standard English?

Reviewer #1: No

Reviewer #3: Yes

Reviewer #4: Yes

6. Review Comments to the Author

Reviewer #1: The manuscript is satisfactory. Writing language is technically appropriate. Statistical analysis is appropriately reported.

Citations of references need to be rechecked. Citation of first reference not found in first paragraph of manuscript.

Reviewer #3: (No Response)

Reviewer #4: PONE-D-21-19745 Reliability of COVID-19 data: An evaluation and reflection PLOS ONE

The article is extremely interesting and relevant. The reviews carried out were adequate. Now, the manuscript technically sound, and do the data support the conclusions.

The revision made was adequate and now we have a intelligible english. Also, The readability of the manuscript improved. The aim is now clear for me.

Somes important questions:

1. Was there any eligibility criteria for selection of volunteers as well as sources included in the study. If Yes; the criteria's can be listed?

2. Finally there is no information regarding ethics approval. Considering that patient information, covid cases and deaths are being recorded, there should be information about the research ethics committee.

7. PLOS authors have the option to publish the peer review history of their article (what does this mean?). If published, this will include your full peer review and any attached files.

Reviewer #1: No

Reviewer #3: No

Reviewer #4: **Yes: **Vicente Sperb Antonello

---

## [Author Response · Author response to Decision Letter 1]

3 Dec 2021

Reviewer 1: Citations of references need to be rechecked. Citation of first reference not found in first paragraph of manuscript.

Response: The citation of the first reference is on the title page of the manuscript, as this is when the COVID-19 data project is first mentioned.

Reviewer 4: Was there any eligibility criteria for selection of volunteers as well as sources included in the study. If Yes; the criteria's can be listed?

Response: For both the selection of volunteers and sources included in the study there was selection criteria. Regarding sources used, these were chosen based on public availability of the datasets and collection methods, along with being cited and used by a variety of organizations such as the CDC and Google. This is stated on page 4 of the manuscript. A more detailed description of eligibility criteria for volunteers has now been included within the “Data Collection Methods” section of the manuscript on page 4. Volunteers were sourced from universities across the United States and were eligible to partake in this project if they had interest or experience in a public health-related field.

Reviewer 4: Finally there is no information regarding ethics approval. Considering that patient information, covid cases and deaths are being recorded, there should be information about the research ethics committee.

Response: This study only used publicly available data from media sources such as New York Times, USA Facts, and John Hopkins University. Publicly available data from county and state public health departments were also used. Seeing as aggregate data is used without providing any identifying information, the data is exempt from IRB approval. Furthermore, the authors are not affiliated with any academic institution. In short, because the data used was public, no ethics approval was sought or obtained.

---

## [Editor Report · Decision Letter 2]

13 Dec 2021

Reliability of COVID-19 data: An evaluation and reflection

PONE-D-21-19745R2

Dear Dr. Schmitt,

We’re pleased to inform you that your manuscript has been judged scientifically suitable for publication and will be formally accepted for publication once it meets all outstanding technical requirements.

Kind regards,

Jagdish Khubchandani, MBBS, PhD

Academic Editor

PLOS ONE

---

## [Editor Report · Acceptance letter]

20 Jan 2022

PONE-D-21-19745R2 

Reliability of COVID-19 data:
An evaluation and reflection 

Dear Dr. Schmitt:

I'm pleased to inform you that your manuscript has been deemed suitable for publication in PLOS ONE. Congratulations! Your manuscript is now with our production department. 

Kind regards, 

on behalf of

Dr. Jagdish Khubchandani 

Academic Editor

PLOS ONE